# Stem Cell Signaling Pathways in the Small Intestine

**DOI:** 10.3390/ijms21062032

**Published:** 2020-03-16

**Authors:** Toshio Takahashi, Akira Shiraishi

**Affiliations:** Suntory Foundation for Life Sciences, Bioorganic Research Institute, Kyoto 619-0284, Japan; shiraishi@sunbor.or.jp

**Keywords:** intestinal stem cell (ISC), organoid, Wnt, Notch-Delta signaling, Hippo signaling, Eph receptor, ephrin, acetylcholine (ACh)

## Abstract

The ability of stem cells to divide and differentiate is necessary for tissue repair and homeostasis. Appropriate spatial and temporal mechanisms are needed. Local intercellular signaling increases expression of specific genes that mediate and maintain differentiation. Diffusible signaling molecules provide concentration-dependent induction of specific patterns of cell types or regions. Differentiation of adjacent cells, on the other hand, requires cell–cell contact and subsequent signaling. These two types of signals work together to allow stem cells to provide what organisms require. The ability to grow organoids has increased our understanding of the cellular and molecular features of small “niches” that modulate stem cell function in various organs, including the small intestine.

## 1. Introduction

The intestinal epithelium undergoes rapid and continuous self-renewal, cell commitment, and cell differentiation along the crypt-villus axis throughout postnatal life. Cheng and Leblond [1] were the first to describe crypt base columnar cells at the crypt bottom that interact with Paneth cells. Paneth cells are intestinal epithelial cells that control stem cells via production of bactericidal products [2] and expression of critical niche signals such as epidermal growth factor, transforming growth factor-α, Wnt3, and delta-like ligand 4 (Dll4), which is the ligand for Notch [3]. Stem cells present at the crypt bottom produce a temporary group of undifferentiated cells that divide rapidly while moving toward the intestinal lumen (Figure 1A) [4]. Subepithelial mesenchymal cells are close to crypts (Figure 1). During migration, these cells differentiate into cells of various lineages; they cease division and differentiate when they reach the top of the crypts (Figure 1A) [4]. Goblet cells, enteroendocrine cells, and absorptive cells, which are also intestinal epithelial cells, move toward the gut lumen, and the Paneth cells move to the crypt bottom (Figure 1A) [5,6]. Intestinal stem cells (ISCs) comprise two functionally distinct populations. Marked leucine-rich repeat-containing G-protein coupled receptor 5 (Lgr5)^+^ cells persist for the lifetime in mice, whereas their progeny includes all differentiated cell lineages of the epithelium (Figure 1B) [7]. Thus, Lgr5^+^ cells represent cycling and long-lived multipotent stem cells. In contrast, B-cell-specific Moloney murine leukemia virus integration site 1 (Bmi1) marks quiescent ISCs, which are located at position +4 from the base of the crypt (Figure 1B). Bmi1^+^ cells are insensitive to Wnt signaling, contribute weakly to homeostatic regeneration, and are resistant to injury from high-dose radiation [8].

Both development of the small intestine and homeostasis of the adult gut require canonical Wnt signaling involving β-catenin and the T-cell factor/lymphoid-enhancer-binding factor family of transcription factors [10,11,12,13,14]. Proliferation in the crypts is compromised in mice lacking canonical Wnt pathway molecules [15]. The central role of Wnt signaling is highlighted by Wnt-dependent expression of numerous ISC markers, including *Lgr5* [16]. Beyond its role in maintaining ISCs, Wnt signaling mediates secretory cell fate decision. Specifically, Wnt signaling plays a role in Paneth cell differentiation [17,18], and overexpression of the Wnt inhibitor dickkopf 1 leads to loss of all secretory cells [19].

Homeostasis of ISCs requires active Notch receptor signaling, as shown with lineage tracing in murine genetic models [20,21] and pharmacological models. Epithelial cell effects have been examined in intestinal organoid cultures [22], which exclude possible influences from Notch signaling in immune, neuronal, and mesenchymal cells. Intestinal organoids grow rapidly and show stable differentiation and are thus useful for nonmutational and genetic experiments [23,24,25]. Overall, organoids are expected to provide critical insight into human and murine ISC signaling regulation.

Erythropoietin-producing hepatocellular carcinoma cell (Eph) receptors and Eph receptor interacting proteins (ephrins) are major players in morphogenesis, where they establish and maintain the organization of cell types or regional domains within tissues [26,27,28]. Eph receptors and ephrins are divided into two classes, in which EphA receptors bind to glycosyl phosphatidyl inositol (GPI) moiety-anchored ephrinAs, and EphB receptors bind to transmembrane ephrinBs [28,29,30]. Multiple tissues and developmental processes involve cross-talk between Wnt and Eph/ephrin signaling pathways [31,32]. Eph/ephrin-mediated cross-talk between epithelial cells controls Wnt signaling and hence spatial regulation of cells located in the crypt-villus axis [4,33]. Cell migration is mediated by Eph receptors and ephrin ligands in many instances [34,35,36]. Eph receptors and ephrins, which are both membrane-bound, only interact through direct cell–cell contact, and signaling through these molecules is bi-directional. Signaling in the receptor-expressing cell is known as forward signaling, and signaling in the ligand-expressing cell is called reverse signaling [27,34,35,37].

The Hippo pathway and its effectors, yes-associated protein (YAP) and yorkie, play a role in intestinal regeneration following tissue injury in mice and *Drosophila melanogaster*, respectively. Activation of the Hippo pathway blocks YAP activity in ISCs via phosphorylation, which induces YAP to remain in the cytoplasm where it interacts with β-catenin and blocks canonical Wnt signaling [38]. Conflicting studies have described the importance of YAP in ISC proliferation and intestinal repair in multiple animal models and experimental paradigms [38]. A recent study [39] established intestinal organoids from single cells. The ability of the intestine to regenerate, which is mediated by transient YAP1 activation, was retained in these organoids.

This review will discuss how ISC division, differentiation, and homeostasis are controlled, how stem cell signaling affects the cell’s position in the crypt-villus axis, and the importance of these observations in development.

## 2. The ISC Niche

### 2.1. R-Spondin-LGR Signaling

Wnt signaling is an evolutionarily conserved pathway that plays a pivotal role in stem cell control. The pathway is tightly controlled not only by ligands and receptors but also by various activators and inhibitors [40]. The ligands, Wnt3, -6, and -9b, are secreted mainly from epithelial cells including Paneth cells, whereas Wnt2b, -4, and -5b are secreted from mesenchymal cells in the intestine [41]. Wnt3 or Paneth cell ablation in vivo does not affect intestinal homeostasis [42,43], indicating that excess biological interactions among ISCs, Paneth cells, and mesenchymal cells maintain the ISC niche via Wnt ligands.

Wnt signaling engages two transmembrane receptor classes, Frizzed (Fz)-type 7-transmembrane proteins and lipoprotein receptor-related proteins 5 and 6 (LRP5/6) (Figure 2). Kazanskaya et al. [44] first identified *Xenopus laevis* R-spondin2 (Rspo2) as a secreted activator of Wnt signaling and showed that Rspo2 is regulated by Wnts and directly activates Wnt signaling. *Xenopus Rspo2* is coexpressed with Wnts in a variety of tissues and can be ectopically activated by Wnt signaling [44]. Similarly, mouse *Rspo1* expression is downregulated in mouse *Wnt1* and *Wnt3a* mutants [45]. As Rsop2 is a secreted protein, Rspo2 may function extracellularly at the level of receptor–ligand interactions during Wnt signaling.

The R-spondin protein family is composed of four human paralogs (R-spondin1–4) (RSpo1–4), each of which contains a leading signal peptide, two cysteine-rich, furin-like domains, and one thrombospondin type 1 domain [44,45]. Using a transgenic mouse model, Kim et al. [46] identified a human gene, *R-spondin1*, with potent and specific proliferative effects on intestinal crypt cells. Examination of human R-spondin1 (hRSpo1) in mouse knock-in chimeras revealed a substantial increase in the diameter, length, and weight of the small intestine [46]. Furthermore, the small intestine of *hRSpo1*-knock-in chimeras revealed a marked, diffuse thickening of the mucosa and crypt epithelial hyperplasia, and a greatly expanded zone of proliferating cells [46]. At that time, the authors speculated that hRSpo1-mediated signaling is not completely dependent on the canonical Wnt/Frizzed pathway, although hRSpo1 may require a distinct frizzled receptor complex [46]. They also emphasized that identification of receptors for R-spondins is urgently necessary to understand the biology of this class of activating ligands [46].

The LGR5 protein is a member of the LGR family. LGR proteins form membrane-spanning structures with large extracellular domains that contain leucine-rich repeats and mediate ligand binding. LGR proteins also contain short cytoplasmic domains that bind heterotrimeric G proteins. The LGR family proteins can be divided into three main groups: types A, B, and C [7]. Type B proteins including, LGR4, LGR5, and LGR6, were considered orphan receptors for more than a decade, but in 2012, the secreted Wnt agonists R-spondins (RSpo1–4) were identified as endogenous ligands of these receptors, revealing the crucial role of LGR proteins in stem cell homeostasis in the gastrointestinal tract [16,47,48,49,50,51]. Gene knockout approaches have shown that LGRs 4–6 play multiple roles during embryonic development and adult tissue homeostasis [52]. The discovery of LGR stem cell markers and their R-spondin ligands has had a major beneficial impact on our understanding of stem cell biology in a range of rapidly renewing tissues [53].

### 2.2. Epidermal Growth Factor (EGF) Signaling

Epidermal growth factor (EGF) is a potent stimulator of proliferation of ISCs and intestinal epithelial cells throughout the crypt-villus axis upon engagement of the EGF family of receptor tyrosine kinases (ErbBs) [54]. Thus, EGF is used as a crucial component of intestinal organoid culture [22]. As ErbB family ligands and receptors are highly expressed within the ISC niche (Figure 2), these endogenous regulators control the pathway in the ISC compartment. Accordingly, Wong et al. [55] revealed that leucine-rich repeats and immunoglobulin-like domain protein 1 (Lrig1), a negative feedback regulator of the ErbB receptor family [56,57,58], is highly expressed by ISCs and controls the size of the ISC niche by regulating the amplitude of growth factor signaling. Their findings position ErbB activation as a strong inductive signal of ISC proliferation.

Bae et al. [59] investigated the relationship between Hippo and other signaling pathways including EGF signaling and the role of Mps one binder (MOB) kinase activator 1A/1B (MOB1A/B) in intestinal homeostasis. MOB1A/B is a core component of the Hippo signaling pathway, which plays a crucial role in cell proliferation, apoptosis, differentiation, and development. MOB1A/B binds to and activates large tumor suppressor 1 and 2 (LATS1/2) kinase, followed by phosphorylation of YAP. Bae et al. discovered that loss of MOB1A/B in intestinal epithelial cells reduces the expression of ISC niche factors including EGF [59]. However, other EGF receptor (EGFR) ligands, including amphiregulin (Areg) and epiregulin (Ereg), are markedly upregulated in intestinal epithelial cells of MOB1A/B-deficient mice [59]. Areg and Ereg transcription can be induced by YAP or transcriptional co-activator with PDZ-binding motif (TAZ) activity [60,61], and thus, their results support the notion that EGF signaling interacts with Hippo signaling. How EGF signaling functions in ISC maintenance and how Hippo signaling interacts with EGF signaling in intestinal epithelial cells remain to be determined.

### 2.3. Bone Morphogenetic Protein (BMP) Signaling

BMP signaling is active in the villus compartment (Figure 2). When BMP signaling is inhibited by Noggin (a BMP antagonist), crypt-like structures appear along the flank of the villi [62], indicating that BMP inhibition creates a crypt-permissive environment. Negative cross-talk is present between Wnt and BMP/transforming growth factor (TGF)-β signaling in the intestine. BMP2 and -4 are expressed in mature epithelial cells and mesenchymal cells [62,63]. In contrast, Noggin is expressed in the crypt region to create a BMP-low niche for ISCs. Suppression of BMP signaling induces expansion of stem and progenitor cells with increasing Wnt activity and eventually leads to development of intestinal polyposis [64]. Accordingly, Noggins such as R-spondin also constitute a key growth factor in the intestinal organoid culture medium cocktail and can be replaced by coculturing with mesenchymal cells [65]. Components of TGF-β signaling are localized in differentiated epithelial cells [66]. BMP/TGF-β-induced mothers against decapentaplegic homolog (Smad) signaling plays roles in differentiation of diverse epithelial cells, and inhibition of Smad leads to stem cell hyperplasia in humans and mice [67]. In particular, activation of BMP receptors by BMP leads to complexes between Smad1/5/8 and Smad4 to repress stemness genes in ISC nuclei [68].

A core component of MOB1A/B in the Hippo signaling pathway coordinates with BMP/TGF-β signaling, the activation of which results in inhibition of Wnt activity in the crypt region and loss of intestinal epithelial homeostasis [59]. This finding is important for understanding Wnt suppression mechanisms by Hippo and BMP/TGF-β signaling in intestinal epithelial cells and for determining which signals associate and interact with each other to influence intestinal epithelial homeostasis.

### 2.4. Notch Signaling in ISC Homeostasis

Notch receptors are single-pass transmembrane heterodimers [69,70,71,72]. Mammals express four Notch receptors (Notch1-4) encoded by different genes (Figure 3A) [71]. Five ligands for the Notch receptor, Dll1, -3, and -4 and Jagged-1 and -2, which are transmembrane Delta/Serrate/Lag2 family members, bind and activate the receptor (Figure 3A) [71,72]. Activated Notch is cleaved in several places, which leads to release of the Notch intracellular domain (NICD) fragment into the cytoplasm (Figure 3A) [72]. In the intestine, the initial cleavage event excising the extracellular receptor domain occurs via the cell surface sheddase, a disintegrin and metalloproteinase 10 (also known as ADAM10) (Figure 3A) [73]. Signals via NICD are transmitted to the nucleus following intramembrane cleavage by the γ-secretase complex (Figure 3A) [74]. NICD binds to the DNA-binding protein, recombination signal binding protein for immunoglobulin kappa J region (also known as RBPJ, CSL-CBF1, suppressor of hairless, Lag2), which changes a transcriptional repressor complex to an activator complex in cooperation with Mastermind (Figure 3A) [74]. This active complex induces transcription of various genes such as Hairy and enhancer of split (Hes) family members [69,70]. Hes1 in turn represses expression of the mouse atonal homolog 1 (Atoh1) transcription factor that is crucial for entry into the secretory lineage [75,76,77].

Notch signaling modulates small intestinal homeostasis via control of *Lgr5*^+^ ISCs and via induction of absorptive cells. Notch signaling induces cells to differentiate into one of two cell types. In the absence of Notch signaling, undifferentiated cells become secretory cells (i.e., enteroendocrine, goblet, and Paneth cells). In the presence of Notch signaling, cells differentiate into enterocytes (Figure 3B) [74]. Secretory cell fate is induced when Notch signaling is blocked by Atoh1 (Figure 3B) [78]. Atoh1 is a critical transcriptional activator that completely induces secretory cell differentiation, as shown by ablation and activation studies in murine genetic models (Figure 3B) [75,76,79]. The key Notch effector that regulates *Atoh1* expression is thought to be the Notch target gene, *Hes1*. For instance, combined loss of both Dll1 and Hes1 results in increased secretory cell types [21,80,81]. Robinson et al. [82] recently reported that the poxvirus and zinc finger transcription factor, Kaiso, regulates the secretory cell lineage in coordination with Notch signaling in intestinal-specific Kaiso-overexpressing mice. In the mice, Kaiso inhibits the expression of Notch1 and Dll1. Consequently, Kaiso-mediated repression of the genes promotes the increase in secretory cells [82]. This finding reveals a novel signaling pathway via Kaiso in regulating Notch-mediated small intestinal homeostasis.

Disruptions in mature secretory cell homeostasis are present in murine models with intestinal-specific gene deletion. The transcription factor Neurogenin3 plays a role in differentiation of enteroendocrine cells, as shown by deletion and overexpression experiments in mice (Figure 3B) [83,84,85]. The transcription repressor, growth factor independent 1 (Gfi1), which is a zinc-finger protein family member, functions downstream of Atoh1 in intestinal secretory lineage differentiation (Figure 3B). *Gfi1*-null mice lack Paneth cells [86]. Another transcription factor, sterile alpha motif pointed domain containing ETS transcription factor (Spdef), when overexpressed in the small intestine, causes the expansion of goblet cells and a corresponding reduction in Paneth cells [87]. Thus, differentiation of goblet/Paneth progenitor cells into mature goblet cells is also regulated by transcription factors (Figure 3B). The colon is devoid of Paneth cells. Paneth-like goblet cells may serve the function of Paneth cells in the colon [88]. Interestingly, the transcript levels of several factors involved in goblet cell differentiation, including Atoh1, Gfi1, Spdef, and E74-like ETS transcription factor 3, are lower in colonic epithelial cells derived from Krüppel-like factor 4 mutant mice compared to controls [89]. These genes may be key players in directing the fate of goblet/Paneth progenitor cells toward mature goblet or Paneth cells.

In the small intestine, maintenance of stem cells and appropriate differentiation into various cell lineages require Notch and Wnt signaling. Previous studies have examined these pathways individually, but how Notch and Wnt signaling work together to maintain ISCs and modulate cell fate remains unclear. The ability to decrease Wnt signaling and thus maintain ISCs is a main function of Notch signaling [90]. Inhibition of Notch induces *Lgr5*^+^ ISCs to differentiate into secretory cells, thus depleting the stem cell pool [90], an observation that is coincident with increased Wnt pathway activity and differentiation into secretory cells [16,17,18,19]. Wnt and Notch signaling likely interact in the biology of many types of stem cells.

### 2.5. Eph/Ephrin Signaling in ISC Homeostasis

Eph receptors compose a large subfamily of receptor tyrosine kinases (Figure 4A). Nine EphA (EphA1–8 and EphA10) and five EphB (EphB1–4 and EphB6) receptors have been identified in humans [91]. Eph ligands are also divided into two types of ephrins according to their structure and binding affinity for EphA or EphB. EphrinAs (A1–A6) are GPI-linked to the plasma membrane, and ephrinBs (B1–B3) are transmembrane proteins with a short cytoplasmic tail (Figure 4A) [91]. EphA receptors generally bind to ephrinA ligands, and EphBs receptors generally bind to ephrinB ligands, although some cross-binding to members of the other type occurs [92].

Increasing evidence supports the roles of EphB/ephrinB signaling in intestinal crypts. Wnt positively regulates cell proliferation and expression of EphB2 and EphB3 [4]. Receptor-positive cells are found in intervillus pockets in the small intestine of neonatal mice and in crypts in the adult small intestine [4]. Conversely, Wnt signaling represses expression of ephrinB1 and ephrinB2 in differentiated cells, and consequently, the distribution of the ligands is largely complementary to EphB receptors [4]. In adult mice, proliferating undifferentiated cells are positive for EphB2, and Paneth cells preferentially express EphB3 (Figure 4B). Double knockout of EphB2 and EphB3 results in complementary expression of EphB receptors and ephrinB ligands to sustain the normal organization of *Lgr5*^+^ ISCs, Paneth cells, and dividing uncommitted cells in the base and differentiated cells in the apex [4,93]. EphB2 and EphB3 also control division of uncommitted cells and re-entry of quiescent cells into the cell cycle [94]. Collectively, in the presence of an increasing gradient of EphB2 receptors, ephrinB ligand-positive precursors are organized from top to bottom in crypts according to a reverse gradient of EphrinB1 and B2 expression (Figure 4B).

### 2.6. The Coordinated Activities of Nicotinic Acetylcholine Receptors (nAChRs), Wnt, and Hippo Signaling in ISC Homeostasis

Cholinergic transmission via acetylcholine (ACh) produces excitatory potentials in postsynaptic cells and is a major excitatory transmission pathway in the enteric nervous system [7]. ACh-mediated neurotransmission in the enteric nervous system is well-understood, but much less is known about non-neuronal ACh, which may modulate intestinal function. Experiments with crypt-villus organoids suggest that ACh is made in intestinal epithelial cells and plays a role in cell division and differentiation of *Lgr5*^+^ ISCs in the small intestine by binding to muscarinic ACh receptors (mAChRs) [24].

Two principal forms of ACh receptors are expressed: mAChRs and nAChRs. Experiments with nicotine and mecamylamine, which is an anti-nicotinic agent, on organoid growth and gene expression in the intestine have shown that endogenous ACh, acting via nAChRs, increases cell division and differentiation and likely modulates interactions between nAChR signaling and other types of signaling [95]. Differential expression of Wnt5a, which activates β-catenin-independent pathways, is critical for downstream nAChR signaling, as shown by our RNA-Seq experiments following stimulation with nicotine and mecamylamine [95]. nAChRs are composed of several types of subunits. Different combinations of subunits, which are variably expressed in neuronal and non-neuronal systems, produce a wide array of physiological and pharmacological effects [96,97]. nAChR subunit-specific antibodies demonstrated the presence of the α2/β4 subtype in the crypt region, suggesting important functions, such as regulation of ISC division and differentiation [95]. We also reported the co-expression of α2 and β4 subunits in Paneth cells [95]. Thus, regulation of ISCs in crypts in the normal adult mouse is associated with increased Wnt5a expression via nAChRs. Endogenous ACh binds to α2/β4 nAChRs in Paneth cells, followed by increased Wnt5a expression (Figure 5A). Wnt binds to Frizzled receptors, which induce noncanonical Wnt signaling, finally leading to increased division and differentiation of *Lgr5*^+^ ISCs (Figure 5A).

Another network that regulates stem cells, the Hippo pathway, is upregulated after treatment with nicotine and downregulated by mecamylamine (Figure 5B,C) [95]. This result reveals that non-neuronal ACh signaling via the α2/β4 nAChR subtype is upstream of the Hippo pathway. The Hippo signaling cascade is composed of highly conserved kinases such as mammalian Ste-like kinase (*hippo* ortholog) and large tumor suppressor kinase (*warts* ortholog), as well as the downstream transcription coactivators, YAP and transcriptional co-activator with a PDZ-binding domain; mammalian homolog of Drosophila *yorkie*. This pathway is critical for maintenance of tissues and organ size, which occurs via regulation of tissue-specific stem cells. Many upstream events initiate the cascade, including mechanical detection of cell density and signaling through G-protein-coupled receptors that regulate Hippo pathway activity [38]. Gestational nicotine treatment has many effects on cell adhesion, which in turn affects expression of the neurexin, immunoglobulin, cadherin, and adhesion G-protein-coupled receptor superfamilies in limbic regions of the brain [98]. Non-neuronal cholinergic systems probably functionally and independently regulate or control cell functions in the mouse intestine.

## 3. Intestinal Organoid Development In Vitro

Organoids are structures that include several cell types that have differentiated from stem cells or organ progenitors. These cells self-organize via spatially restricted differentiation into various lineages and migration in a way that recapitulates the in vivo situation [99]. The mouse intestinal organoid is a typical example that first grows as a single-layered epithelium that then self-organizes into domains that are structurally similar to the in vivo intestinal crypt-villus. These organoids contain the various cell types present in the intestine that surround a cystic lumen (Figure 6) [7].

Isolated crypts containing *Lgr5*^+^ ISCs require Matrigel, EGF, Noggin, and R-spondin 1 in serum-free medium [7]. The above three factors represent the minimal and essential stem cell maintenance factors as a cocktail for growth (Figure 6A). In addition to Matrigel support, several groups have developed non-Matrigel strategies for the culture of ISCs and intestinal organoids. Gjorevski and Lutolf [100] describe a protocol for the generation of well-defined matrices. These matrices comprise a polyethylene glycol hydrogel backbone functionalized with minimal adhesion cues, including RGD (Arg-Gly-Asp), which is sufficient for ISC expansion, and laminin-111, which is required for organoid formation. Recently, Capeling et al. [101] demonstrated that alginate, a minimally supportive hydrogel with no inherent cell instructive properties, supports growth of human intestinal organoids (HIOs) in vitro and leads to HIO epithelial differentiation that is virtually indistinguishable from Matrigel-grown HIOs. Culturing single ISCs is inefficient, with less than 2% plating efficiency, whereas up to half of ISC–Paneth cell doublets form organoids in vitro [3,7]. As Paneth cells are believed to create a niche environment and determine the future crypt site, emergence of a single Paneth cell is important for organoid development derived from a single ISC (Figure 6A).

Although researchers studying ISC biology extensively use the intestinal organoid culture system, how single ISCs give rise to a cell population with the capability of self-organization and which transcriptional program cells use are unclear. Serra and coworkers [39] clearly demonstrated that intestinal organoid formation is a regenerative process that relies on transient YAP1 activation, following a regeneration process (Figure 6B). YAP1 is a mechanosensing nuclear effector of the Hippo pathway and regulates organ growth, regeneration, and tumorigenesis [102,103]. Organoid development requires temporary cell–cell variability in YAP localization, which then leads to Notch and Dll1 lateral inhibition between the 8- and 16-cell stage (Figure 6B) [39]. These events lead to the first instance of symmetry breaking in intestinal organoid growth and initiate differentiation of the first Paneth cell (Figure 6B) [39]. Thus, single cells can induce self-organization that leads to multicellular asymmetric structures like embryos.

## 4. Conclusions

Stem cells can divide extensively and have the ability to produce committed cells during early mammalian development. Early in development, stem cells need to be able to divide and differentiate into various lineages in response to external signals. Stem cells have specialized internal mechanisms that allow these activities, and hence, these cells are initially present when niches can sequester, maintain, and regulate undifferentiated embryonic cells [104]. Adult stem cells in tissues also have tremendous potential. They are maintained in a steady state in which each division typically produces one replacement stem cell and one tissue cell with no apparent limit [7].

The basic architecture of the intestinal tract includes a simple epithelial layer. The small intestinal villus and its associated epithelium include enterocytes as the main cell type and secretory cell types, whereas the crypt is the site of cell division and hence the origin of all epithelial components. A complex gradient of factors maintains ISC stemness and proliferation along the crypt-villus axis. In the crypts, Paneth cells express factors that promote stem cell growth, including Wnt3a, Dll1/4, and EGF [3]. In addition, several factors, such as Wnt2b [42], Rspo1 [16,46,105], Gremlin1, Gremlin2 (BMP antagonists), and Chordin [106,107], produced by mesenchymal cells, play an essential role in the maintenance of ISCs.

The extracellular matrix (ECM) is a dynamic and complex environment characterized by biophysical, mechanical, and biochemical properties specific for each tissue and is able to regulate cell behavior. The ECM is an essential player in the stem cell niche, because it can directly or indirectly modulate the maintenance, proliferation, self-renewal, and differentiation of stem cells. Several ECM molecules have regulatory functions for different types of stem cells. ECM proteins and integrin receptors interact to keep ISCs within the niche [108]. Exploration of the ECM as an influencer of ISC niche formation and maintenance is ongoing [109,110,111,112]. Engineered biomaterials that mimic the in vivo characteristics of the stem cell niche will provide suitable in vitro tools for dissecting the different roles of the ECM.

The mesenchymally derived basement membrane dynamically controls morphogenesis, cell differentiation, and polarity, while also providing the structural basis for villi, crypts, and the microvasculature of the lamina propria so that crucially, tissue morphology is preserved in the absence of epithelium. Interestingly, the biomechanical work of Balbi and Ciarletta [113] supports the influence of material properties of the developing intestinal epithelium on how the villi are formed, which is closely linked to many of the signaling gradients discussed here. They demonstrated that the emergence of intestinal villi in embryos is triggered by differential growth between the epithelial mucosa and mesenchymal tissues [113]. Their proposed morpho-elastic model highlights that both the geometric and mechanical properties of the epithelial mucosa strongly influence the formation of pre-villous structures in embryos, providing useful suggestions for interpreting the dynamics of villus morphogenesis in living organisms [113]. This model may provide guidance for artificial formation of villus structures using complex signaling pathways.

Intrinsic stem cell capabilities mediate establishment of complex tissues during development and/or repair in murine [22] and *D. melanogaster* [114] intestines. The existence of an intrinsic process for tissue generation and repair is clear. What is less clear is whether the stem cell or the niche comes first. The Wnt pathway is present in the simplest multicellular organisms and is thus evolutionarily old [115,116]. In the earliest metazoans, Wnt appears to be an ancestral signal that breaks symmetry to divide a previously symmetric embryo into anterior and posterior domains, allowing evolutionary emergence of asymmetric organisms [117,118]. Individual cells produce genetic and nongenetic heterogeneity, which leads to specialized cell behavior [119,120,121]. Thus, single cells can break the symmetry of a population by affecting differentiation in relation to other identical cells [122]. Indeed, cell-to-cell variability in intestinal organoid development occurs owing to the combined effects of three factors: Wnt, Yap1, and Notch/Dll1 activation (Figure 6B).

Recently, Ayyaz and coworkers [123] identified a revival stem cell, termed revSC, with the use of single-cell RNA sequencing to profile the regenerating mouse intestine. revSCs are typically rare and are characterized by high expression of clusterin [123]. Radiation-induced intestinal damage, specific depletion of *Lgr5*^+^ ISCs, or treatment with dextran sodium sulfate causes revSCs, which are critical for intestinal regeneration, to temporarily express YAP1 and restore the *Lgr5*^+^ ISC pool [123]. YAP1-dependent, injury-induced revSC division may be critical for tissue repair following damage.

## Figures and Tables

**Figure 1 ijms-21-02032-f001:**
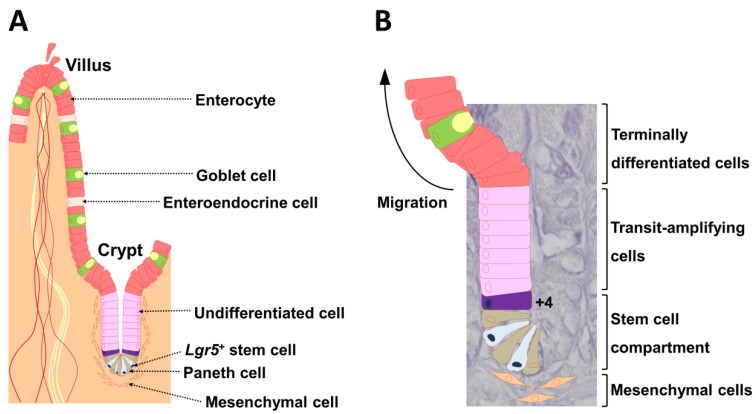
The crypt-villus unit of the small intestinal epithelium. (**A**) The epithelium is mainly composed of enterocytes, goblet cells, enteroendocrine cells, and Paneth cells. *Lgr5*^+^ stem cells reside at the very bottom of the crypt (position +1) and are usually flanked on both sides by Paneth cells. Subepithelial mesenchymal cells are localized near *Lgr5*^+^ stem cells. (**B**) The hierarchical organization is illustrated in a 4′,6-diamidino-2-phenylindole (DAPI)-stained crypt. Daughter cells differentiate into functionally discrete populations of intestinal epithelial cells as they migrate up from the crypt, and are extruded into the gut lumen as they reach the villus tip in the small intestine. The cell located at position +4 from the base of the crypt also undergoes self-renewal and gives rise to all differentiated cell lineages of the small intestinal epithelium. Lgr5: leucine-rich, repeat-containing G-protein coupled receptor 5 [9].

**Figure 2 ijms-21-02032-f002:**
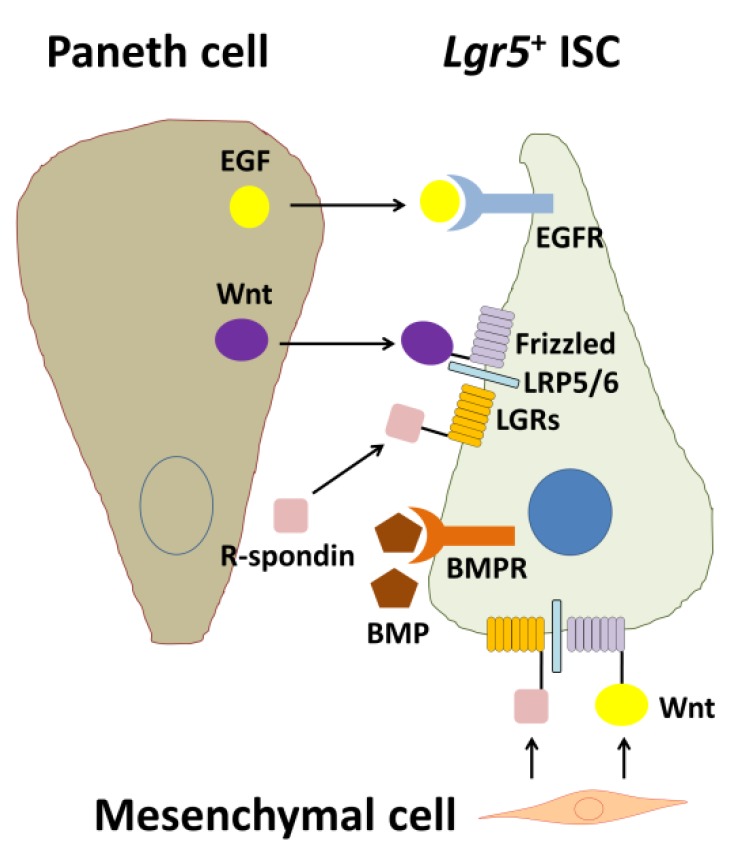
Biological interactions among an *Lgr5*^+^ intestinal stem cell (ISC), a Paneth cell, and a mesenchymal cell. EGF and Wnt are secreted from the Paneth cell and are essential for stemness of the *Lgr5*^+^ ISC. In contrast, BMP negatively regulates the stemness. For full Wnt activation, R-spondin-LGR signaling is required. In addition, Wnt and R-spondins, which are factors produced by mesenchymal cells, are also essential for maintenance of *Lgr5*^+^ ISCs. EGF: epidermal growth factor, EGFR: EGF receptor, BMP: bone morphogenetic protein, BMPR: BMP receptor, LRP5/6: lipoprotein receptor-related protein 5/6, LGRs: leucine-rich repeat-containing G-protein coupled receptors, Lgr5: leucine-rich repeat-containing G-protein coupled receptor 5 [9].

**Figure 3 ijms-21-02032-f003:**
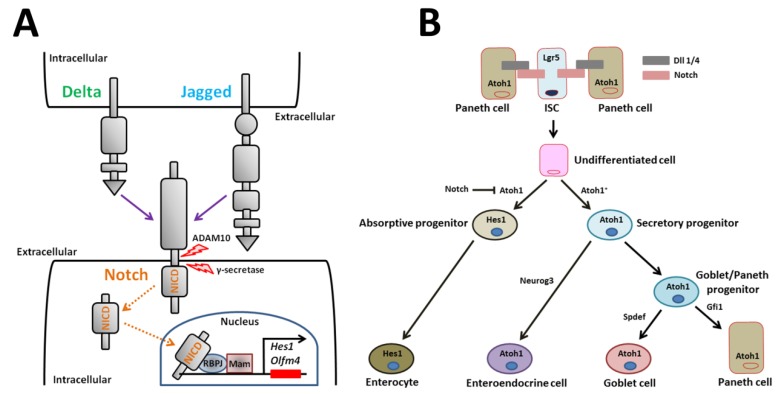
Notch signaling and model for Notch regulation of small intestinal epithelial cell differentiation. (**A**) Transmission of Notch signaling involves engagement of two cells. The signal-sending cell expresses the Notch ligands, Delta and Jagged. The signal-receiving cell expresses Notch receptors. After ligand/receptor engagement, the Notch receptor is proteolytically cleaved. The cleaved NICD translocates to the nucleus where it recruits a transcriptional coactivator complex that activates downstream transcription of Notch target genes such as *Hes1*. (**B**) Notch signaling represses the bHLH transcriptional regulator, Atoh1, to form absorptive enterocytes. Atoh1^+^ secretory progenitor cells are fated towards three mature secretory cell types. Neurog3 induces enteroendocrine cell differentiation. Spdef regulates Goblet cell differentiation. Gfi1 regulates Paneth cell differentiation. ADAM10: a disintegrin and metalloproteinase 10, NICD: Notch intracellular domain, RBPJ: recombination signal binding protein for immunoglobulin kappa J region, Mam: mastermind, Hes1: Hairy and enhancer of split 1, Olfm4: olfactomedin 4, Atoh1: atonal homolog 1, Lgr5: leucine-rich repeat-containing G-protein coupled receptor 5, ISC: intestinal stem cell, Dll: delta-like ligand, Neurog3: neurogenin 3, Gfi1: growth factor independent 1, Spdef: sterile alpha motif pointed domain containing ETS transcription factor.

**Figure 4 ijms-21-02032-f004:**
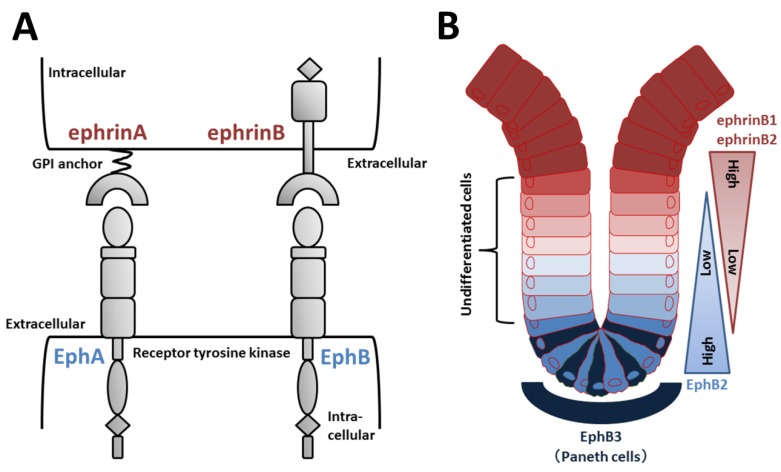
Eph/ephrin structure and regulation of EphB/ephrinB signaling in intestinal crypts. (**A**) Domain organization of Eph receptors and ephrin ligands. EphA receptors typically bind to ephrinA (GPI-anchored) ligands, and EphB receptors bind to ephrinB ligands. (**B**) EphB2-positive stem and undifferentiated cells are located near the bottom of intestinal crypts, whereas ephrinB1/B2-positive differentiated epithelial cells are concentrated at the crypt-villus boundary. In the small intestine, EphB3-expressing Paneth cells are found at the crypt bottom.

**Figure 5 ijms-21-02032-f005:**
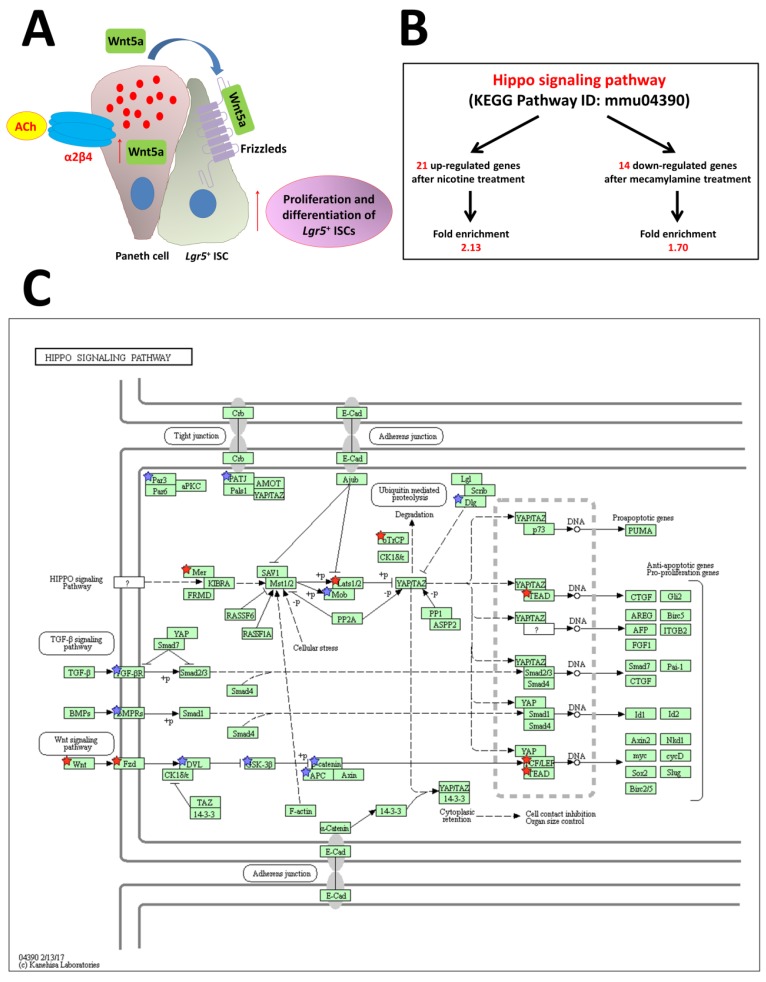
Model for the proposed role of nAChR signaling pathways in ISC function. (**A**) Non-neuronal ACh activates the nicotinic receptor α2β4, which is localized in Paneth cells, to modulate the expression level of *Wnt5a* [9]. Wnt then modulates Wnt pathway activity through frizzled receptors. Eventually, proliferation and differentiation of stem cells are enhanced. (**B**) The stem cell regulation network, the Hippo pathway, is regulated by nAChR signaling. (**C**) Hippo signaling pathway (mmu04390) maps derived from Database for Annotation, Visualization, and Integrated Discovery (DAVID) analysis. The genes with red stars were upregulated by nicotine and downregulated by mecamylamine. Additionally, the genes with blue stars were upregulated by nicotine but not downregulated by mecamylamine. ACh: acetylcholine, Lgr5: leucine-rich repeat-containing G-protein-coupled receptor 5 [95].

**Figure 6 ijms-21-02032-f006:**
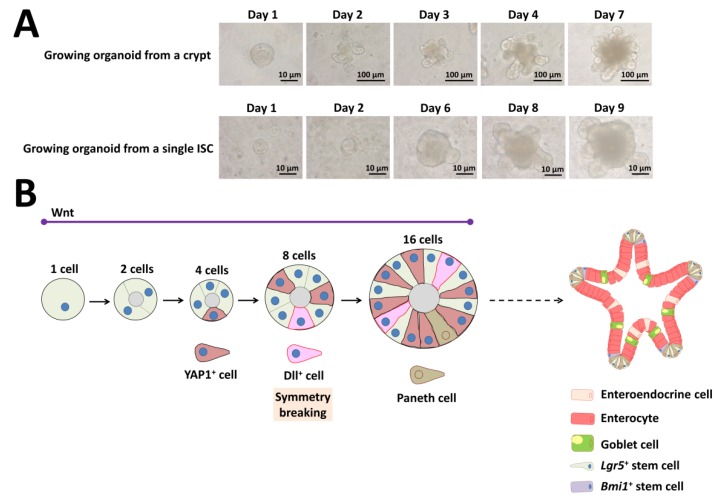
Self-organization and symmetry breaking in intestinal organoid development. (**A**) Representative images of a growing organoid derived from a crypt or from a single ISC. (**B**) Model of organoid development and symmetry breaking. YAP1: yes-associated protein 1, Dll: delta-like ligand, Lgr5: leucine-rich repeat-containing G-protein-coupled receptor 5, Bmi1: B-cell-specific Moloney murine leukemia virus integration site 1.

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
