# Peer review of "Stem Cell Signaling Pathways in the Small Intestine"

_ijms, 2020, doi:10.3390/ijms21062032_

Round 1

Reviewer 1 Report

I appreciate the authors' work in this review of stem cells in the gut. The manuscript is now ready for publication.

Author Response

1) Reviewer's comments and suggestions for authors

I appreciate the authors' work in this review of stem cells in the gut. The manuscript is now ready for publication.

Responses to reviewer's comments

I am very pleased to hear that our manuscript is worth to be ready for publication. Thank you very much for taking out your precious time for reviewing our manuscript.

Reviewer 2 Report

The authors are commended on a significantly improved manuscript. This reviewer feels that this manuscript now describes the key signalling pathways governing intestinal stem cell identity and its differentiation in a comprehensive way. 

As minor revision I request that is made clear in title and abstract (and with a few sentences in the manuscript) that this review pertains to the small intestine in particular. The architecture and signalling cell types in the large intestine/colon are very different.

Author Response

2) Reviewer's comments and suggestions for authors

The authors are commended on a significantly improved manuscript. This reviewer feels that this manuscript now describes the key signalling pathways governing intestinal stem cell identity and its differentiation in a comprehensive way.

As minor revision I request that is made clear in title and abstract (and with a few sentences in the manuscript) that this review pertains to the small intestine in particular. The architecture and signalling cell types in the large intestine/colon are very different.

Responses to reviewer's comments

Thank you very much for taking out your precious time for reviewing our manuscript. I understand that our manuscript especially pertains to the small intestine. We agree with the reviewer's comment. Thus, to clearly show it in the title and the abstract, we changed the word "gut" to "small intestine". Additionally, we clarified to be small intestine in a few sentences in our manuscript as follows.

Title

Stem cell signaling pathways in the gut -> Stem cell signaling pathways in the small intestine

Abstract

Page1, line15: including the gut. -> including the small intestine.

Manuscript

Page6, line198: (Figure3 legend) Notch signaling and model for Notch regulation of small intestinal epithelial cell differentiation.

Page6, line213: Notch signaling modulates small intestinal homeostasis...

Page6, line226: ...via Kaiso in regulating Notch-mediated small intestinal homeostasis.

Page7, line234-235: ... when overexpressed in the small intestine, ...

This manuscript is a resubmission of an earlier submission. The following is a list of the peer review reports and author responses from that submission.

Round 1

Reviewer 1 Report

Overview: The authors have provided a well written and well organized review of the recent literature surrounding intestinal stem cell homeostasis and the figures are very helpful to visualize each section. This will serve as a strong contribution to the field and deserves publication after addressing a few minor considerations.

Comments:

Introduction

Last paragraph of Intro. Re-write to passive tense to fit rest of the review style. Should read something like

“This review will discuss how ISC division, differentiation, and homeostasis are controlled, how stem cell signaling affects the cell’s position in the crypt-villus axis, and the importance of these observations in development.”

Intestinal organoid development in vitro

Line 233-234: In addition to Matrigel, several groups have developed non-Matrigel strategies to get Lgr5+ cells (e.g., Kristi Anseth’s group has made ISC organoid formation fine-tuned and predictable). I suggest including this range of work as well or comment on how there is still exploration going into the extracellular matrix and how it influences ISC niche formation/maintenance: hanging drop: 1007/s10616-018-0194-8 collagen: https://doi.org/10.1155/2019/8472712 alginate: https://doi.org/10.1016/j.stemcr.2018.12.001 PEG/laminin: 10.1038/nprot.2017.095 Review of ECM in ISC niche: 1155/2017/7970385

Conclusion

It may be worth also mentioning the physical ISC aspects since they are not extensively reviewed here (or even the above ECM comments to here). For example, the biomechanical work of Balbi and Ciarletta (1098/rsif.2013.0109) supports the influence of material properties of the developing intestine epithelial on how the villi are formed (which is closely linked to many of the signaling gradients discussed).

Author Response

Overview: The authors have provided a well written and well organized review of the recent literature surrounding intestinal stem cell homeostasis and the figures are very helpful to visualize each section. This will serve as a strong contribution to the field and deserves publication after addressing a few minor considerations.

Comments:

Introduction

Comment 1: Last paragraph of Intro. Re-write to passive tense to fit rest of the review style. Should read something like

“This review will discuss how ISC division, differentiation, and homeostasis are controlled, how stem cell signaling affects the cell’s position in the crypt-villus axis, and the importance of these observations in development.”

Answer: Thank you very much for your helpful suggestion concerning the last paragraph of Intro. Accordingly, we re-wrote to passive tense to fit rest of the review style following your sentence.

Last paragraph of Intro: This review will discuss how ISC division, differentiation, and homeostasis are controlled, how stem cell signaling affects the cell’s position in the crypt-villus axis, and the importance of these observations in development.

Intestinal organoid development in vitro

Comment: Line 233-234: In addition to Matrigel, several groups have developed non-Matrigel strategies to get Lgr5+ cells (e.g., Kristi Anseth’s group has made ISC organoid formation fine-tuned and predictable). I suggest including this range of work as well or comment on how there is still exploration going into the extracellular matrix and how it influences ISC niche formation/maintenance: hanging drop: 1007/s10616-018-0194-8 collagen: https://doi.org/10.1155/2019/8472712 alginate: https://doi.org/10.1016/j.stemcr.2018.12.001 PEG/laminin: 10.1038/nprot.2017.095 Review of ECM in ISC niche: 1155/2017/7970385

Answer: As I did not cover non-Matrigel strategies for ISC organoid growth and extracellular matrix for ISC niche, reviewer's suggestion is fruitful for me. Thus, I added the following sentences and references.

Line 236~: In addition to Matrigel support, several groups have developed non-Matrigel strategies for the culture of ISCs and intestinal organoids. Gjorevski and Lutolf [73] describe a protocol for the generation of well-defined matrices. These matrices comprise a polyethylene glycol hydrogel backbone functionalized with minimal adhesion cues including RGD (Arg-Gly-Asp), which is sufficient for ISC expansion, and laminin-111, which is required for organoid formation. Recently, Capeling et al. [74] demonstrate that alginate, a minimally supportive hydrogel with no inherent cell instructive properties, support human intestinal organoids (HIOs) growth in vitro and leads to HIO epithelial differentiation that is virtually indistinguishable from Matrigel-grown HIOs.

(New references)

[73] Gjorevski, N.; Lutolf, M.P. Synthesis and characterization of well-defined hydrogel matrices and their application to intestinal stem cell and organoid culture. Nature Protoc. 2017, 12, 2263-2274.

[74] Capeling, M.M.; Czerwinski, M.; Huang, S.; Tsai, Y-H.; Wu, A.; Nagy, M.S.; Juliar, B.; Sundaram, N.; Song, Y.; Han, W.M.; et al. Nonadhesive alginate hydrogels support growth of pluripotent stem cell-derived intestinal organoids. Stem Cell Reports 2019, 12, 381-394.

Conclusions (ECM) (line 265~): Extracellular matrix (ECM) is a dynamic and complex environment characterized by biophysical, mechanical and biochemical properties specific for each tissue and able to regulate cell behavior. ECM represents an essential player in stem cell niche, since it can directly or indirectly modulate the maintenance, proliferation, self-renewal and differentiation of stem cells. Several ECM molecules play regulatory functions for different types of stem cells. ECM proteins and integrin receptors interact to keep ISCs within the niche [90]. There is still exploration going into the ECM for influences of ISC niche formation and maintenance [91-94]. Engineered biomaterials able to mimic the in vivo characteristics of stem cell niche will provide suitable in vitro tools for dissecting the different roles exerted by the ECM.

(New references)

[90] Lin, G.; Zhang, X.; Ren, J.; Pang, Z.; Wang, C.; Xu, N.; Xi, R. Integrin signaling is required for maintenance and proliferation of intestinal stem cells in Drosophila. Dev. Biol. 2013, 377, 177–187.

[91] Schultz, K.M.; Kyburz, K.A.; Anseth, K.S. Measuring dynamic cell-material interactions and remodeling during 3D human mesenchymal stem cell migration in hydrogels. Proc. Natl. Acad. Sci. USA 2015, 112, E3757–E3764.

[92] Panek, M.; Grabacka, M.; Pierzchalska, M. The formation of intestinal organoids in a hanging drop culture. Cytotechnology 2018, 70, 1085–1095.

[93] Jee, J.H.; Lee, D.H.; Ko, J.; Hahn, S.; Jeong, S.Y.; Kim, H.K.; Park, E.; Choi, S.Y.; Jeong, S.; Lee, J.W.; et al. Development of collagen-based 3D matrix for gastrointestinal tract-derived organoid culture. Stem Cells Int. 2019, 2019, 8472712.

[94] Meran, L.; Baulies, A.; Li, V.S.W. Intestinal stem cell niche: The extracellular matrix and cellular components. Stem Cells Int. 2017, 2017, 7970385.

Conclusion

Comment: It may be worth also mentioning the physical ISC aspects since they are not extensively reviewed here (or even the above ECM comments to here). For example, the biomechanical work of Balbi and Ciarletta (1098/rsif.2013.0109) supports the influence of material properties of the developing intestine epithelial on how the villi are formed (which is closely linked to many of the signaling gradients discussed).

Answer: As I also did not cover the physical ISC aspects for building the small intestinal villi, reviewer's suggestion is helpful for me. Thus, I added the following new paragraph and references.

Line 265~: The basic architecture of the intestinal tract includes a simple epithelial layer. The small intestinal villus and its associated epithelium includes enterocytes as the main cell type and secretory cell types, while the crypt is the site of cell division and hence the origin of all epithelial components.

(New paragraph) The mesenchymally-derived basement membrane dynamically controls morphogenesis, cell differentiation and polarity, while also providing the structural basis for villi, crypts and the microvasculature of the lamina propria so that tissue morphology, crucially, is also preserved in the absence of epithelium. Interestingly, the biomechanical work of Balbi and Ciarletta [95] supports the influence of material properties of the developing intestinal epithelium on how the villi are formed, which is closely linked to many of the signaling gradients discussed here. They demonstrate that the emergence of intestinal villi in embryos is triggered by a differential growth between the epithelial mucosa and the mesenchymal tissues [95]. The proposed morpho-elastic model highlights that both the geometrical and the mechanical properties of the epithelial mucosa strongly influence the formation of pre-villous structures in embryos, providing useful suggestions for interpreting the dynamics of villus morphogenesis in living organisms [95]. The model may give us a hint for artificial formation of villus structure using complex signaling pathways.

(New reference)

[95] Balbi, V.; Ciarletta, P. Morpho-elasticity of intestinal villi. J. R. Soc. Interface 2013, 10, 20130109.

Reviewer 2 Report

The review article by Takhashi et al aims to give an overview about “stem cell signalling pathways in the gut” that control intestinal stem cell maintenance and differentiation.

In the authors’ own words: “I will discuss how ISC division, differentiation, and homeostasis are controlled, and how stem cell signalling affects the cell’s position in the crypt-villus axis.”

The abstract hints at an exciting review that summaries current knowledge about soluble factors and direct cell interactions controlling these processes. Unfortunately, the review falls very short of these lofty aims. It is characterised by poor use of the English language, an unclear structure and highly deficient content. In order for an article to cover this topic the authors should have given a clear overview of the cellular composition of the epithelium and any supporting structures. Indeed, the primary niche component that the authors mention are Paneth cells, but they are only part of the niche, together with the underlying mesenchyme. As such deletion of Paneth cells does not compromise ISC maintenance and therefore describing and summarizing current knowledge about the mesenchymal niche component would have been as critical.

Furthermore, the auhors' focus almost exclusively on a very few signalling pathways while only mentioning in passing the most critical ones, like canonical Wnt signalling (R-Spondin), BMP signalling (Noggin) and EGF signalling, even though they are the key pathways that maintain intestinal organoids in vitro and ISC indentity in general. Discussion of some pathways that can bias differentiation and regeneration outcomes will be of limited use to readers considering the bigger picture view is so deficient.

Author Response

The review article by Takhashi et al aims to give an overview about “stem cell signalling pathways in the gut” that control intestinal stem cell maintenance and differentiation.

Comment: In the authors’ own words: “I will discuss how ISC division, differentiation, and homeostasis are controlled, and how stem cell signalling affects the cell’s position in the crypt-villus axis.”

Answer: We re-wrote to passive tense to fit rest of the review style as follows.

(Introduction) pg2, lines 73 and 74: This review will discuss how ISC division, differentiation, and homeostasis are controlled, how stem cell signaling affects the cell’s position in the crypt-villus axis, and the importance of these observations in development.

Comment: The abstract hints at an exciting review that summaries current knowledge about soluble factors and direct cell interactions controlling these processes. Unfortunately, the review falls very short of these lofty aims. It is characterised by poor use of the English language, an unclear structure and highly deficient content. In order for an article to cover this topic the authors should have given a clear overview of the cellular composition of the epithelium and any supporting structures.

Answer: To give readers a clear overview of the cellular composition of the epithelium and any suppirting structures, I added a new figure (Figure 1).

Figure legend (New Figure 1): The crypt-villus unit of the small intestinal epithelium. (A) The epithelium is mainly composed of enterocytes, goblet cells, enteroendocrine cells, and Paneth cells. Lgr5+ stem cells reside at the very bottom of the crypt (position +1), and are usually flanked on both sides by Paneth cells. Subepithelial mesenchymal cells are localized near Lgr5+ stem cells. (B) The hierarchical organization is illustrated in a DAPI (4',6-diamidino-2-phenylindole)-stained crypt. Daughter cells differentiate into functionally discrete populations of intestinal epithelial cells as they migrate up from the crypt, and are extruded into the gut lumen as they reach the villus tip in the small intestine. The cell located at position +4 from the base of the crypt also undergoes self-renewal and gives rise to all differentiated cell lineages of the small intestinal epithelium. Lgr5: leucine-rich repeat-containing G-protein coupled receptor 5.

Comment: Indeed, the primary niche component that the authors mention are Paneth cells, but they are only part of the niche, together with the underlying mesenchyme. As such deletion of Paneth cells does not compromise ISC maintenance and therefore describing and summarizing current knowledge about the mesenchymal niche component would have been as critical.

Answer: According to the reviewer's comment, we described and summarized current knowledge about mesenchymal niche component in conclusions as follows. And also, we added new references.

Conclusions (line 265~): A complex gradient of factors maintains ISC stemness and proliferation along the crypt-villus axis. In the crypts, Paneth cells express factors that promote stem cells growth, including Wnt3a, Dll1/4, and EGF [3]. In addition, a number of factors produced by mesenchymal cells have an essential role in the maintenance of ISCs such as Wnt2b [85], Rspo1 [15,86,87], and Gremlin1 (Grem1; BMP antagonist) [88,89].

(New references)

[85] Farin, H.F.; Van Es, J.H.; Clevers, H. Redundant sources of Wnt regulate intestinal stem cells and promote formation of Paneth cells. Gastroenterology 2012, 143, 1518–1529.e7.

[86] Kim, K.A.; Kakitani, M.; Zhao, J.; Oshima, T.; Tang, T.; Binnerts, M.; Liu, Y.; Boyle, B.; Park, E.; Emtage, P.; et al. Mitogenic influence of human R-spondin1 on the intestinal epithelium. Science 2005, 309, 1256–1259.

[87] Ootani, A.; Li, X.; Sangiorgi, E.; Ho, Q.T.; Ueno, H.; Toda, S.; Sugihara, H.; Fujimoto, K.; Weissman, I.L.; Capecchi, M.R.; et al. Sustained in vitro intestinal epithelial culture within a Wnt-dependent stem cell niche. Nat. Med. 2009, 15, 701–706.

[88] Kosinski, C.; Li, V.S.; Chan, A.S.; Zhang, J.; Ho, C. Tsui, W.Y.; Chan, T.L.; Mifflin, R.C.; Powell, D.W.; Yuen, S.T.; et al. Gene expression patterns of human colon tops and basal crypts and BMP antagonists as intestinal stem cell niche factors. Proc. Natl. Acad. Sci. USA 2007, 104, 15418–15423.

[89] Hsu, D.R.; Economides, A.N.; Wang, X.; Eimon, P.M.; Harland, R.M. The Xenopus dorsalizing factor Gremlin identifies a novel family of secreted proteins that antagonize BMP activities. Mol. Cell 1998, 1, 673–683.

Comment: Furthermore, the auhors' focus almost exclusively on a very few signalling pathways while only mentioning in passing the most critical ones, like canonical Wnt signalling (R-Spondin), BMP signalling (Noggin) and EGF signalling, even though they are the key pathways that maintain intestinal organoids in vitro and ISC indentity in general. Discussion of some pathways that can bias differentiation and regeneration outcomes will be of limited use to readers considering the bigger picture view is so deficient.

Answer: We agree with the reviewer's comment. It is worth discussing some pathways that can bias differentiation and regeneration outcomes. Accordingly, we will discuss extracellular matrix (ECM) and the physical ISC aspects for ISC niche.

Conclusions (ECM) (line 265~): Extracellular matrix (ECM) is a dynamic and complex environment characterized by biophysical, mechanical and biochemical properties specific for each tissue and able to regulate cell behavior. ECM represents an essential player in stem cell niche, since it can directly or indirectly modulate the maintenance, proliferation, self-renewal and differentiation of stem cells. Several ECM molecules play regulatory functions for different types of stem cells. ECM proteins and integrin receptors interact to keep ISCs within the niche [90]. There is still exploration going into the ECM for influences of ISC niche formation and maintenance [91-94]. Engineered biomaterials able to mimic the in vivo characteristics of stem cell niche will provide suitable in vitro tools for dissecting the different roles exerted by the ECM.

(New references)

[90] Lin, G.; Zhang, X.; Ren, J.; Pang, Z.; Wang, C.; Xu, N.; Xi, R. Integrin signaling is required for maintenance and proliferation of intestinal stem cells in Drosophila. Dev. Biol. 2013, 377, 177–187.

[91] Schultz, K.M.; Kyburz, K.A.; Anseth, K.S. Measuring dynamic cell-material interactions and remodeling during 3D human mesenchymal stem cell migration in hydrogels. Proc. Natl. Acad. Sci. USA 2015, 112, E3757–E3764.

[92] Panek, M.; Grabacka, M.; Pierzchalska, M. The formation of intestinal organoids in a hanging drop culture. Cytotechnology 2018, 70, 1085–1095.

[93] Jee, J.H.; Lee, D.H.; Ko, J.; Hahn, S.; Jeong, S.Y.; Kim, H.K.; Park, E.; Choi, S.Y.; Jeong, S.; Lee, J.W.; et al. Development of collagen-based 3D matrix for gastrointestinal tract-derived organoid culture. Stem Cells Int. 2019, 2019, 8472712.

[94] Meran, L.; Baulies, A.; Li, V.S.W. Intestinal stem cell niche: The extracellular matrix and cellular components. Stem Cells Int. 2017, 2017, 7970385.

Conclusions (the physical ISC aspects): The mesenchymally-derived basement membrane dynamically controls morphogenesis, cell differentiation and polarity, while also providing the structural basis for villi, crypts and the microvasculature of the lamina propria so that tissue morphology, crucially, is also preserved in the absence of epithelium. Interestingly, the biomechanical work of Balbi and Ciarletta [95] supports the influence of material properties of the developing intestinal epithelium on how the villi are formed, which is closely linked to many of the signaling gradients discussed here. They demonstrate that the emergence of intestinal villi in embryos is triggered by a differential growth between the epithelial mucosa and the mesenchymal tissues [95]. The proposed morpho-elastic model highlights that both the geometrical and the mechanical properties of the epithelial mucosa strongly influence the formation of pre-villous structures in embryos, providing useful suggestions for interpreting the dynamics of villus morphogenesis in living organisms [95]. The model may give us a hint for artificial formation of villus structure using complex signaling pathways.

(New reference)

[95] Balbi, V.; Ciarletta, P. Morpho-elasticity of intestinal villi. J. R. Soc. Interface 2013, 10, 20130109.

Reviewer 3 Report

The short review on the “Stem cell signaling pathways in the gut” by Toshio Takahashi and Akira Shiraishi provides a concise overview on intestinal stem cells and underlying signaling pathways in the gut. Overall this is a nicely written review content-wise and summarizes the topic well. There are no major concerns.

Minor comments- grammar, syntax need attention throughout the article. For instance, in pg 2, lines 73 and 74 In this review, “I will discuss how ISC division, differentiation, and homeostasis are controlled, and how stem cell signaling affects the cell’s position in the crypt-villus axis. I also describe the …." I should be replaced with we.

Also, in this review, topical developments that describe functional roles of Wnts and Rspo ligands in the maintenance as well as the differentiation of intestinal crypt stem cell niche were not sufficiently described.

A number of recent publications (for eg, PMC5641471) have shown mechanistic insights into this key signaling pathway ISC biology.

The review would become much more attractive if authors can succinctly summarize few recent advancements in this important signaling pathway.

Author Response

The short review on the “Stem cell signaling pathways in the gut” by Toshio Takahashi and Akira Shiraishi provides a concise overview on intestinal stem cells and underlying signaling pathways in the gut. Overall this is a nicely written review content-wise and summarizes the topic well. There are no major concerns. Minor comments- grammar, syntax need attention throughout the article.

Comment: For instance, in pg 2, lines 73 and 74 In this review, “I will discuss how ISC division, differentiation, and homeostasis are controlled, and how stem cell signaling affects the cell’s position in the crypt-villus axis. I also describe the …." I should be replaced with.

Answer: Thank you very much for your helpful suggestion concerning the last paragraph of Introduction (pg2, lines 73 and 74). Accordingly, we will re-write to passive tense to fit rest of the review style as follows.

pg2, lines 73 and 74: This review will discuss how ISC division, differentiation, and homeostasis are controlled, how stem cell signaling affects the cell’s position in the crypt-villus axis, and the importance of these observations in development.

Comment: Also, in this review, topical developments that describe functional roles of Wnts and Rspo ligands in the maintenance as well as the differentiation of intestinal crypt stem cell niche were not sufficiently described.

Answer: As I did not cover functional roles of Wnts and Rspo in the maintenance as well as the differentiation of intestinal crypt stem cell niche, reviewer's suggestion is fruitful for me. Thus, I added the following new paragraph and references.

pg7, line240~: LGR5 protein belongs to the LGR family. LGR proteins form membrane-spanning structures with large extracellular domains that contain leucine-rich repeats and mediate ligand binding. LGR proteins also contain short cytoplasmic domains that bind heterotrimeric G proteins The LGR family proteins can be divided into three main groups: types A, B, and C [7]. Type B including LGR4, LGR5, and LGR6 were considered orphan receptors for more than a decade, but in 2012, the secreted Wnt agonists R-spondins (RSpo1–4) were identified as endogenous ligands of these receptors, revealing the crucial role of LGR proteins in stem cell homeostasis in the gastrointestinal tract [15, 75-79]. Gene knockout approaches have shown that LGR 4–6 play multiple roles during embryonic development and adult tissue homeostasis [80]. The discovery of LGR stem cell markers and their R-spondin ligands has had a major beneficial impact on our understanding of stem cell biology in a range of rapidly renewing tissues [81].

(New references)

[75] Carmon, K.S.; Gong, X.; Lin, Q.; Thomas, A.; Liu, Q. R-spondins function as ligands of the orphan receptors LGR4 and LGR5 to regulate Wnt/β-catenin signaling. Proc. Natl. Acad. Sci. USA 2011, 108, 11452–11457.

[76] Carmon, K.S.; Lin, Q.; Gong, X.; Thomas, A.; Liu, Q. LGR5 interacts and co-internalizes with Wnt receptors to modulate Wnt/β-catenin signaling. Mol. Cell Biol. 2012, 32, 2054–2064.

[77] Glinka, A.; Dolde, C.; Kirsch, N.; Huang, Y.L.; Kazanskaya, O.; Ingelfinger, D.; Boutros, M.; Cruciat, C.M.; Niehrs, C. LGR4 and LGR5 are R-spondin receptors mediating Wnt/β-catenin and Wnt/PCP signaling. EMBO Rep. 2011, 12, 1055–1061.

[78] Ruffner, H.; Sprunger, J.; Charlat, O.; Leighton-Davies, J.; Grosshans, B.; Salathe, A.; Zietzling, S.; Beck, V.; Therier, M.; Isken, A.; et al. R-spondin potentiates Wnt/β-catenin signaling through orphan receptors LGR4 and LGR5. PLoS One 2012, 7, e40976.

[79] Gong, X.; Carmon, K.S.; Lin, Q.; Thomas, A.; Yi, J.; Liu, Q. LGR6 is a high affinity receptor of R-spondins and potentially functions as a tumor suppressor. PLoS One 2012, 7, e37137.

[80] Barker, N.; Clevers, H. Leucine-rich repeat-containing G-protein-coupled receptors as markers of adult stem cells. Gastroenterology 2010, 138, 1681–1696.

[81] Barker, N.; Tan, S.; Clevers, H. Lgr proteins in epithelial stem cell biology. Development 2013, 140, 2484–2494.

Comment: A number of recent publications (for eg, PMC5641471) have shown mechanistic insights into this key signaling pathway ISC biology. The review would become much more attractive if authors can succinctly summarize few recent advancements in this important signaling pathway.

Answer: We agree with the reviewer's comment. It is worth mentioning mechanistic insights into this key signaling pathway of ISC biology. As I also did not cover the physical ISC aspects, reviewer's suggestion is helpful for me. Thus, I added the following new paragraph and references.

Line 265~: The basic architecture of the intestinal tract includes a simple epithelial layer. The small intestinal villus and its associated epithelium includes enterocytes as the main cell type and secretory cell types, while the crypt is the site of cell division and hence the origin of all epithelial components.

(New paragraph) The mesenchymally-derived basement membrane dynamically controls morphogenesis, cell differentiation and polarity, while also providing the structural basis for villi, crypts and the microvasculature of the lamina propria so that tissue morphology, crucially, is also preserved in the absence of epithelium. Interestingly, the biomechanical work of Balbi and Ciarletta [95] supports the influence of material properties of the developing intestinal epithelium on how the villi are formed, which is closely linked to many of the signaling gradients discussed here. They demonstrate that the emergence of intestinal villi in embryos is triggered by a differential growth between the epithelial mucosa and the mesenchymal tissues [95]. The proposed morpho-elastic model highlights that both the geometrical and the mechanical properties of the epithelial mucosa strongly influence the formation of pre-villous structures in embryos, providing useful suggestions for interpreting the dynamics of villus morphogenesis in living organisms [95]. The model may give us a hint for artificial formation of villus structure using complex signaling pathways.

(New reference)

[95] Balbi, V.; Ciarletta, P. Morpho-elasticity of intestinal villi. J. R. Soc. Interface 2013, 10, 20130109.

Round 2

Reviewer 2 Report

Ultimately I stick to my initial assessment and deem the article not helpful for the  scientific community. I realise that I am being overruled by the other 2 reviewers who have different views.  There are dozens of beautifully written reviews about the signally pathways in the intestinal stem cell compartment that provide a concise and accessible overview of how stem cells are maintained in this organ system. There are for example 3 essential signalling molecules that are critical to maintain intestinal stem cells and ex-vivo (as so called organoids). R-spondin, EGF and Noggin. Noggin is mentioned once in the article (with no explanation of how it is acting). EGF is mention twice with little context (other than where secreted from) and certainly no details about its mode of action. From my perspective this review article is lacking in key content and I have doubts it will be of help to the field or received as a useful resource by other scientists. However as I have been over ruled by the other 2 reviewers I genuinely wish the authors, editors and the journal overall only the best moving forward with the article.